# Molecular epidemiology, genetic diversity and antimicrobial resistance of *Staphylococcus aureus* isolated from chicken and pig carcasses, and carcass handlers

Onyinye J. Okorie-Kanu[1]*, Madubuike U. Anyanwu[2], Ekene V. Ezenduka[1], Anthony C. Mgbeahuruike[2], Dipendra Thapaliya[3], Gracen Gerbig[3], Ejike E. Ugwuijem[4], Christian O. Okorie-Kanu[5], Philip Agbowo[1], Solomon Olorunleke[1], John A. Nwanta[1], Kennedy F. Chah[2], Tara C. Smith[3]

1 Department of Veterinary Public Health and Preventive Medicine, University of Nigeria, Nsukka, Enugu State, Nigeria, 2 Microbiology Unit, Department of Veterinary Microbiology and Pathology, University of Nigeria, Nsukka, Enugu State, Nigeria, 3 Department of Biostatistics, Environmental Health Sciences and Epidemiology, Kent State University, Kent, Ohio, United States of America, 4 Department of Microbiology, University of Nigeria, Nsukka, Enugu State, Nigeria, 5 Department of Veterinary Pathology, Michael Okpara University of Agriculture, Umudike, Abia State, Nigeria

☯ These authors contributed equally to this work.
* onyinye.okoro@unn.edu.ng

**Data Availability Statement:** All relevant data are within the paper.

## Abstract

The epidemiology of *Staphylococcus aureus* in food animals, associated products, and their zoonotic potential in Nigeria are poorly understood. This study aimed to provide data on the prevalence, genetic characteristics and antimicrobial resistance of *S. aureus* isolated from chicken and pig carcasses, and persons in contact with the carcasses at slaughterhouses in Nigeria. Surface swabs were collected randomly from 600 chicken and 600 pig carcasses. Nasal swabs were collected from 45 workers in chicken slaughterhouses and 45 pig slaughterhouse workers. *S. aureus* isolates were analyzed by *spa* typing. They were also examined for presence of the Panton-Valentine Leucocidin (PVL) and *mecA* genes, as well as for antimicrobial resistance phenotype. Overall, 53 *S. aureus* isolates were recovered (28 from chicken carcasses, 17 from pig carcasses, 5 from chicken carcass handlers and 3 from pig carcass handlers). Among the isolates, 19 (35.8%) were PVL-positive and 12 (22.6%) carried the *mecA* gene. The 53 isolates belonged to 19 *spa* types. The Based Upon Repeat Pattern (BURP) algorithm separated the isolates into 2 *spa*-clonal complexes (*spa*-CC) and 9 singletons including 2 novel *spa* types (t18345 and t18346). The clonal complexes (CC) detected were CC1, CC5, CC8, CC15, CC88 and CC152. CC15-related isolates represented by *spa* type t084 (32.1%) and CC5 represented by *spa* type t311 (35.3%) predominated among isolates from chicken carcasses/ handlers, and pig carcasses/ handlers, respectively. Multidrug resistance exhibited by all the CC except CC8, was observed among isolates from chicken carcasses (64.3%), pig carcasses (41.2%), handlers of chicken meat (40.0%) and handlers of pork (33.3%). All the CC showed varying degrees of resistance to tetracycline while CC15 and CC5 exhibited the highest resistance to sulphamethoxazole/trimethoprim and erythromycin, respectively. The predominant antimicrobial resistance

**Funding:** This study was sponsored by the authors and partly supported by a grant received by OJO from the University of Nigeria, Nsukka through the "Needs Assessment Intervention fund" for Academic Staff Training and Development. We are also grateful to Smith Emerging Infections Laboratory for releasing materials for the work when the authors' funds and grant could no longer sustain the work.

**Competing interests:** The authors have declared that no competing interests exist.

pattern observed was penicillin-tetracycline-sulphamethoxazole/trimethoprim (PEN-TET-SXT). In conclusion, food animals processed in Enugu State in Southeast Nigeria are potential vehicles for transmission of PVL-positive multiple-drug resistant *S. aureus* and methicillin-resistant *S. aureus* from farm to slaughterhouse and potentially to the human population. Public health intervention programs at pre- and post-slaughter stages should be considered in Nigerian slaughterhouses.

## Introduction

*Staphylococcus aureus* is a commensal of the skin and mucous membranes, especially the anterior nares (nostrils) in humans and animals. It can cause opportunistic infections following trauma of the skin and mucous membranes [1]. There is an increased interest in strains of *S. aureus* because these organisms are associated with a wide variety of zoonotic infections ranging from mild skin infections to life-threatening invasive diseases [2, 3]. In addition, they exhibit resistance to many antibiotics, with methicillin-resistant *S. aureus* (MRSA) strains being resistant to most β-lactam antibiotics [4]. *S. aureus* may also be resistant to other classes of antibiotics such as fluoroquinolones, aminoglycosides and tetracyclines; resistance has also been reported to last-resort drugs for resistant *S. aureus* infections such as vancomycin, a glycopeptide [5] and inducible-clindamycin-resistant *S. aureus* strains are increasingly reported [6]. Infections caused by strains resistant to three or more classes of antimicrobial agents, also known as multidrug-resistant (MDR) strains, are increasingly difficult to treat [7].

MRSA has been identified as a key pathogen in nosocomial (hospital-associated [HA]), community-associated (CA), and livestock-associated (LA) infections [8]. CA strains commonly carry the Panton-Valentine Leukocidin (PVL), a putative virulence factor that induces pore formation in the membranes of cells and is encoded by *lukS* and *lukF* genes [9]. This toxin is common in *S. aureus* identified in Africa [10]. Globally, MRSA constitutes a significant health threat. It was estimated to cause 84,000 invasive infections and 11,000 deaths in United States in 2011 [11], 17,000 blood stream infections and 5,400 deaths in Europe in 2007 [12], and a reported 32% mortality from MRSA bacteraemia cases in Hong Kong [13]. The threat is likely equal or more in developing countries [14]. Thus, the World Health Organization (WHO) recently classified MRSA as "high priority 2 pathogens" that are a threat to the health of humans and animals, and against which new management strategies and research documenting their occurrence in different reservoirs are urgently needed.

Although there is evidence of zoonotic transmission of MRSA, the source(s) of MRSA in humans is an ongoing question, and additional evidence to support animal-to-human transmission is still needed [8]. Thus there is an increased interest in tracking, identifying and understanding the diversity of *S. aureus* in various settings [10]. The identification of bacterial clones with enhanced virulence or increased ability to spread is very important. Molecular epidemiological studies involving animal and human reservoirs are crucial for determining the sources of *S. aureus* and MRSA in an ecological niche, deducing the pathogenic characteristics of the strains, and developing effective control strategies, while the determination of antimicrobial susceptibility of *S. aureus* isolates is important for targeted empirical therapy [10,15]. Currently, PCR-based techniques are commonly used for the characterization of isolates as they are fast and easy to use. Among such techniques, staphylococcal protein A (*spa*) gene typing also known as *spa* typing, is the most promising sequence-based method for the epidemiological studies of *S. aureus* [16]. The *spa* gene, which encodes a surface coat protein known as

Protein A, is conserved among *S. aureus* strains. This gene provides suitable short sequence repeat region (known as the X- region), which contains variable number tandem repeat (VNTRs) that are highly polymorphic and are used as a target for single-locus sequence typing (SLST), popularly known as *spa* typing [17]. Unfortunately, these molecular techniques have not been well-utilized in Nigeria, especially at the animal-human interface.

Elsewhere, there are several studies on molecular characterization of *S. aureus* isolated from pigs and raw pork, chickens and chicken meats, and human handlers [18,19]. In Nigeria, although live pigs and poultry at slaughter or farm are screened for MRSA [20], phenotypic detection based on conventional biochemical tests are used for identification. This testing may not be reliable, especially in Nigeria where biochemical reagents are often purchased from traders. However, only Nworie et al. [21] in Ebonyi State, Southeast Nigeria and Ayeni et al. [22] in Ogun State, in the Southwest used genotypic characterization techniques to detect *S. aureus* isolates from poultry while Odetokun et al. [23] in the Southwest and, Momoh et al. [24] as well as Otalu et al. [25] in the Northcentral also reported the molecular epidemiology of *S. aureus* from food animals and occupationally-exposed humans in Nigeria. Molecular epidemiological data on *S. aureus* contaminating raw meat meant for human consumption and humans with occupational contact are limited in Nigeria. This study was therefore undertaken to investigate the prevalence and genotypes of *S. aureus* in chicken and pig carcasses, and occupationally-exposed humans at slaughterhouses in Enugu State, Southeast Nigeria.

## Materials and methods

### Ethical statement

All procedures used in this study were in accordance with the revised version of the Animals Scientific Procedures Act of 1986 for the care and use of animals for research purposes. Permission to conduct this study was also given by the Research Ethics Committee of the Faculty of Veterinary Medicine, University of Nigeria, Nsukka and the Medical Research Ethics Committee of the University of Nigeria.

### Study area

This study was conducted in Enugu State in Southeast Nigeria. The state is geographically located between latitudes 5˚56' North and 7˚55' North, and longitudes 6˚53' East and 7˚55' East. It is comprised of 3 agricultural zones: Awgu, Enugu, and Nsukka. The 3 zones are made up of 17 local government areas. The population of Enugu state is 3,267,837 [26]. Poultry and pigs are the main sources of animal protein for the Enugu State populace.

### Sampling

This cross-sectional study was conducted from January to August, 2018. In each of the agricultural zones, a slaughterhouse per agricultural zone was selected based on the average slaughter capacity. Each of the selected slaughterhouses was visited once per week. From each agricultural zone, 200 each of freshly processed pigs and chicken carcasses consisting of 10% of total slaughter within the period of the study were selected using a 1 in 4 systematic random sampling technique. After processing, each chicken carcass was dipped in the same container of water for washing while for pig carcasses, each was washed using a separate container of water. These carcasses do not undergo any form of treatment like chilling (as is done in developed countries) before being sold to the general public. Swab samples were collected from 1,200 randomly selected carcasses (600 chicken and 600 pigs) which represent 10% of the total carcasses processed during the sampling period. Sterile swabs were used to swab the surfaces and inner

cavities of the freshly processed carcasses after evisceration. During the visits to each agricultural zone, 15 adult volunteers each in contact with poultry and pig carcasses who gave written informed consent in accordance with the Declaration of Helsinki [27] were randomly selected per agricultural zone. These individuals slaughter the animals and process the carcasses through de-feathering/de-hairing, evisceration, washing and packaging. Nasal swabs were collected from each of the subject volunteers using sterile swabs. A total of 90 nasal swab samples were collected (45 from persons in contact with poultry carcass and 45 in contact with pig carcass), and these represent 5% of the total occupationally-exposed humans sampled during the sampling period. The swab samples were transported with ice packs to the laboratory and processed within one hour after collection for *S. aureus* isolation.

## Bacterial isolation and identification

The swabs were inoculated into 5ml nutrient broth (Oxoid, Bangistoke UK) containing 6.5% NaCl and incubated at 37˚C for 24 hours in ambient air. A loopful of the broth culture was inoculated onto Baird-Parker agar (BPA) (Oxoid, Bangistoke UK) containing egg yolk tellurite (EYT) and incubated at 37˚C for 48 hours in ambient air. Presumptive *S. aureus* isolates (shiny black colonies with clear halos with or without opaque zones) were purified on BPA with EYT at 37˚C for 24 hours. The isolates were Gram stained, subjected to catalase test, slide and tube coagulase tests, *S. aureus* latex agglutination assay (Pastorex Staph-plus, Bio rad) and haemolysis test (inoculating onto Columbia colistin nalidixic acid agar with 5% sheep blood and incubated at 37˚C for 24 hours in ambient air). Isolates phenotypically identified as *S. aureus* were subjected to further species confirmation and molecular characterization at the Smith Emerging Infections laboratory, Kent State University, Ohio.

## Molecular characterization

Genomic DNA was extracted using the Wizard Genomic DNA preparation kit (Promega, Madison WI) following the manufacturer's instructions. Polymerase Chain Reaction (PCR) was done on all the isolates. The presence of *mecA* was determined by PCR using *mecA*-F (5'-AAA ATC GAT GGT AAA GGT TGG C-3') and *mecA*-R (5'-AGT TCT GCA GTA CCG GAT TTG C-3') primers, following previously described protocol [28] while the presence of PVL gene (*lukF*, *lukS*) was determined using *luk*-PV-F (5'-ATC ATT AGG TAA AAT GTC TGG ACA TGA TCC A-3') and *luk*-PV-R (5'-GCA TCA AST GTA TTG GAT AGC AAA AGC-3') primers following previously described procedures [29]. The polymorphic X-region of the *spa* gene was amplified (*spa* typing) from all *S. aureus* isolates using the primers *spa*F (5'-GAA CAA CGT AAC GGC TTC ATC C-3') and *spa*1514R (5'-CAG CAG TAG TGC CGT TTG CCT-3') by adopting previously described methods [17,30]. The *spa* typing technique uses the polymorphic VNTR sequence in the 3' coding region of the *spa* gene. Each new base of the polymorphic repeat found in a strain of *S. aureus* is assigned a unique repeat numeric code known as the repeat succession, which invariably determines the *spa* type (t) of that strain [17,30]. The *spa* types were assigned using Ridom StaphType software (http://spaserver.ridom.de) version 2.2.1 (Ridom GmbH, Wurzburg, Germany). The Based Upon Repeat Pattern (BURP) algorithm implemented by the software was applied to *spa* types to group *S. aureus* isolates, based on their genetic proximity, into larger related genetic clusters known as *spa*-cluster complex (*spa*-CC) [31]. Since excellent concordance between *spa* types and MLST results have been established in previous studies [32], only *spa* typing was done. *spa* types (t) were placed into particular clonal complexes (CC) using information in the Ridom StaphType software. Positive (USA 300, [33]) and negative controls (reaction mixture without DNA template) were used for PVL, *mecA* and *spa* PCR.

## Antimicrobial susceptibility testing (AST)

The isolates were tested for antimicrobial susceptibility (AST) using VITEK 2 System (BioMerieux, Durham, NC) following the manufacturer's instructions and in accordance with the Clinical and Laboratory Standards Institute (CLSI) guidelines [34]. The isolates were tested against a panel of 18 antimicrobial agents belonging to 13 classes: β-lactam: benzylpenicillin (PEN), oxacillin (OXA) and ceftaroline (CFR); aminoglycosides: gentamicin (GEN); fluoroquinolone: ciprofloxacin (CIP), levofloxacin (LEV), and moxifloxacin (MOX); macrolide: erythromycin (ERY); lincosamide: clindamycin (CLI); oxazolidinone: linezolid (LIN); lipopeptide: daptomycin (DPT); glycopeptide: vancomycin (VAN); tetracycline: minocycline (MIN) and tetracycline (TET); glycylcycline: tigecycline (TIG); nitroheterocyclics: nitrofurantoin (NIT); ansamycin: rifampicin (RIF); and folate pathway antagonists: sulphamethoxazole/ trimethoprim (SXT). USA300 was used as a reference strain. Results (minimum inhibitory concentration [MIC]) of the AST were provided and interpreted by the VITEK2 system according to the CLSI guidelines using MIC breakpoints for staphylococci [35]. Intermediately-susceptible isolates were classified as resistant. An isolate resistant to three or more classes of antimicrobial agents, or to methicillin/oxacillin, was considered MDR [36]. Inducible clindamycin resistance was tested using the VITEK2 system.

## Statistical analysis

The frequencies of occurrence of *S. aureus* and resistance of the isolates to antimicrobial agents were entered into Microsoft Excel version 2010 (Microsoft Corporation, Redmond, USA) and subjected to descriptive statistics to determine their percentages. Association between categorical variables was tested using Chi-square (Fisher's exact and Pearson's) test with Graph Pad Prism. Statistical significance was accepted at p< 0.05.

# Results

## Prevalence of *S. aureus* and MRSA

A total of 1,290 samples were examined for the presence of *S. aureus*. The overall prevalence of *S. aureus* was 4.1% (53/1,290; 95% confidence interval [CI] 3.0%–5.2%) see (Table 1). Chicken carcasses had a significantly higher prevalence of *S. aureus* when compared to pig carcass and carcass handlers (P = 0.0256).

The *mecA* gene was detected in 12 (22.6%; 95% CI 11.3%–33.9%) of the 53 *S. aureus* strains identified. Thus, the overall prevalence of methicillin-resistant *S. aureus* (MRSA) based on the detection of the *mecA* gene, was 0.9% (12/1,290; 95% CI 0.41%–1.45%) with prevalence rates of 1.5% (9/600; 95% CI 0.84%–2.16%) in chicken carcasses and 0.5% (3/600; 95% CI 0.12%–0.89%) in pig carcasses, respectively; none of the *S. aureus* isolates from the chicken and pig carcass handlers was positive for the *mecA* gene (Table 1). Even though chicken carcasses had higher prevalence of MRSA when compared to pig carcasses, this was not significant (P = 0.488).

**Table 1. Prevalence of *S. aureus* and MRSA in food animal carcass and carcass handlers.**

| Sample | Samples processed (N) | *S. aureus* (%; 95% CI) | MRSA (%; 95% CI) | MSSA (%; 95% CI) |
|---|---|---|---|---|
| Chicken carcass | 600 | 28 (4.7%; 3.8%–5.6%) | 9 (1.5%; 0.84%–2.16%) | 19 (3.2%; 1.8%–4.6%) |
| Pig carcass | 600 | 17 (2.8%; 1.5%–4.12%) | 3 (0.5%; 0.12%–0.89%) | 14 (2.3%; 1.1%–3.5%) |
| Chicken carcass handlers (Nasal swabs) | 45 | 5 (11.1%; 2.2%–19.8%) | 0 (0.0; 0.0%–0.0%) | 5 (11.1%; 1.9%–20.3%) |
| Pig carcass handlers (Nasal swabs) | 45 | 3 (6.7%; 0.6%–14.0%) | 0 (0.0%; 0.0%–0.0%) | 3 (6.7%; 0.0%–14.0%) |
| Total | 1,290 | 53 (4.1%; 3.0%–5.2%) | 12 (0.9%; 0.4%–1.4%) | 41 (3.2%; 2.2%–4.1%) |

CI = Confidence interval; methicillin resistance in *S. aureus* was based on detection of the *mecA* gene.

## Clustering of isolates by *spa* typing and Based Upon Repeat Pattern (BURP) algorithm analyses

The sequencing of the *spa* genes revealed that among the 53 *S. aureus* isolates, 19 *spa* types were identified, including 4 *S. aureus* isolates (7.5%) that did not match any known sequence. These novel *spa* sequences, with repeat successions 26-16-20-17-12-12-17-16 and 26-12-21-17-13-34-34-34, were submitted to the Ridom SpaServer through the Ridom StaphType web-site and were assigned new *spa* types (t18345 and t18346). The two most predominant *spa* types were t311 (12; 22.6%) and t084 (9; 17.0%), representing 40% of all the isolates detected. The remaining 17 *spa* types each represented 7.5% or less of all isolates. These include t786 (7.5%), t1931 (5.7%), t448 (5.7%), t18345 (5.7%), t085 (3.8%), t2393 (3.8%), t304 (3.8%), t355 (3.8%), t5562 (3.8%), t934 (3.8%), t14223 (1.9%), t18346 (1.9%), t2216 (1.9%), t279 (1.9%), t346 (1.9%), t4690 (1.9%) and t491 (1.9%). The 19 *spa* types were grouped into 6 clonal complexes (CC) (Table 2).

The BURP algorithm is used to group *spa* types into larger genetic clusters known as *spa*-cluster complex (*spa*-CC). It sums up "costs" (a measure of the relatedness based on the *spa* types) to define a founder score for each *spa* type in a *spa*-CC. The founder *spa* type (represented by a blue node) (Fig 1) is the *spa* type with the highest founder score in its *spa*-CC. The BURP analysis of the *spa* types clustered almost half (49%) of the isolates into 2 *spa*-CCs, with 28% of the isolates clustered in *spa*-CC084 and 21% in *spa*-CC448 (Fig 1). Thirty-two percent of all *spa* types were clustered into *spa*-CC084 (founder t084) and 21% in *spa*-CC448 (founder t448). *spa* types that differ from all other *spa* types in the sample by more than four repeats could not reasonably be clustered into a *spa*-CC and were termed singletons. Nine *spa* types (27 isolates) which make up 51% of all the isolates were classified as singletons. They were t304, t311, t355, t934, t1931, t4690, t14223, including the 2 novel *spa* types (t18345 and t18346) (Fig 1).

Isolates with *spa* types belonging to CC5 and CC15 were identified in all sample types processed in this study. Isolates with *spa* types belonging to CC1, CC88 and CC152 were identified in chicken and pig carcasses only and not in carcass handlers.

The 28 isolates from chicken carcasses belonged to 13 *spa* types and 6 major clonal complexes: CC15 (n = 9; 32.1%), C88 (n = 7; 25.0%), CC1 (n = 4, 14.3%), CC5 (n = 3; 10.5%), CC8 and CC152 (n = 2 each; 7.1%), and an undetermined CC (CC-ND) with a new *spa* type t18345 (n = 1; 3.6%). Seventeen isolates from pig carcass belonged to 8 *spa* types and 5 major clonal complexes: CC5 (n = 6, 35.3%), CC88 (n = 4, 23.5%), CC15, CC152 and CC-ND with a new *spa* type, t18345 (n = 2 each, 11.8%) and CC1 (n = 1, 5.9%). Five isolates from chicken carcass handlers belonged to 4 *spa* types and 2 major clonal complexes: CC15 (n = 3; 60.0%), CC5 and CC-ND with a new *spa* type t18346 (n = 1 each; 20.0%) while the 3 isolates from pig carcass handlers belonged to 2 *spa* types in CC5 (n = 2; 66.7%) and CC15 (n = 1, 33.3%) (Table 2).

## Detection of PVL genes among *S. aureus* isolates

Out of 53 *S. aureus* isolates detected, 19 (35.8%) were PVL-positive; these comprised 14 (73.6%) MSSA and 5 (26.3%) MRSA. Eleven (39.3%; 95% CI 21.2%–57.4%) out of the 28 *S. aureus* isolates from chicken were PVL-positive, 4 of which also harbored the *mecA* gene (Table 2). Among the 17 *S. aureus* isolates from pig carcasses, 5 (29.5%, 95% CI 7.8%–51.2%) were PVL-positive, and one harbored *mecA*. Only 1 (33.3%, 95% CI 20.0%–86.6%) PVL-positive strain was detected among the 3 *S. aureus* isolates from pig carcass handlers whereas 2 (40.0%, 95% CI 18.1%–61.9%) of the 5 *S. aureus* isolates from chicken carcass handlers were PVL-positive (Table 2). PVL-positive *S. aureus* (MSSA or MRSA) strains generally occurred more frequently in chicken than in pig carcasses and the human handlers. However, the association between occurrence and source was not significant ($\chi^2 = 0.4958$; $P = 0.9198$).

**Table 2. *spa* cluster complex of *S. aureus* isolates from food animal carcasses and occupationally-exposed persons in Enugu State, Southeast Nigeria.**

| Source (N) | Cluster complex (representative *spa* type) | Number of isolates (%) | *spa* types (N) | MRSA (N, %) *spa* type | MSSA (N, %) *spa* type | N (%) PVL-positive strain |
|---|---|---|---|---|---|---|
| Poultry carcass (28) | CC15 (t084) | 9 (32.1) | t084 (7) t085 (2) | 2 (7.1) t085 (1) t084 (1) | 7 (28.0) t084 (6) t085 (1) | 6 (21.4) MSSA (4) MRSA (2) |
| | CC1 | 4 (14.3) | t1931(2) t934 (2) | 2 (7.1) t1931(2) | 2 (7.1) t934 (2) | 2 (7.1) MRSA |
| | CC5 (t311) | 3 (10.7) | t311 (3) | 2 (7.1) t311 (2) | 1 (3.6) t311 (1) | 0 (0.0) |
| | CC8 (t304) | 2 (7.1) | t304 (2) | 0(0.0) | 2 (7.1) t304 (2) | 0 (0.0) |
| | CC88 | 7 (25.0) | t448 (2) t2393 (1) t5562 (2) t786 (2) | 3 (10.7) t5562 (1) t786 (2) | 4 (14.2) t448 (2) t2393(1) t5562 (1) | 0 (0.0) |
| | CC152 | 2 (7.1) | t355 (1) t4690 (1) | 0 (0.0) | 2 (7.1) t355 t4690 | 2 (7.1) MSSA |
| | CC-ND | 1 (3.6) | t18345 (1) | 0 (0.0) | 1 (3.6) t18345 (1) | 1 (3.6) MSSA |
| **Total** | | **28 (52.8)** | | **9 (32.1)** | **19 (67.9)** | **11 (39.3) MSSA = 7 (63.6) MRSA = 4 (36.4)** |
| Pig carcass (17) | CC5(t311) | 6 (35.3) | t311 (6) | 0 (0.0) | 6 (35.3) t311 (6) | 1 (5.9), MSSA |
| | CC1(t1931) | 1 (5.9) | t1931 (1) | 1 (5.9) t1931 (1) | 0 (0.0) | 1 (5.9), MRSA |
| | CC15 | 2 (11.8) | t2216 (1) t346 (1) | 0(0.0) | 2 (11.8) t2216 (1) t346 (1) | 0 (0.0) |
| | CC88 | 4 (14.3) | t448 (2) t786 (2) | 2 (11.8) t786 (2) | 2 (11.8) t448 | 0 (0.0) |
| | CC152 (t355) | 2 (11.8) | t355 (2) | 0 (0.0) | 2 (11.8) t304 (2) | 1 (5.9),MSSA |
| | CC-ND (t18345) | 2 (11.8) | t18345(2) | 0 (0.0) | 2 (11.8) t18345 (2) | 2 (11.8), MSSA |
| Total | | | | 3 (17.6) | 14 (82.4) | 5 (29.4) MSSA = 4 (80.0) MRSA = 1 (20.0) |
| Chicken carcass handlers (5) | CC15 (t084) | 3 (40.0) | t084 t491 | 0 (0.0) | 3 (40.0) t084 (2) t491 (1) | 1 (20.0) MSSA |
| | CC5 (t311) | 1 (20.0) | t311 (1) | 0(0.0) | 1 (20.0) t311 | 0 (0.0) |
| | CC-ND (t18346) | 1 (20.0) | t18346 (1) | 0(0.0) | 1 (20.0) t18346 (1) | 1 (20.0) MSSA |
| Total | | **5 (100.0)** | | **0 (0.0)** | **5 (100.0)** | **2 (40.0) MSSA = 2 (40.0) No MRSA = 0** |
| Pig carcass handlers (3) | CC5 (t311) | 2 (66.7) | t311 (2) | 0 (0.0) | 2 (66.7) t311 | 0 (0.0) |
| | CC15 (t279) | 1 (33.3) | t279 (1) | 0 (0.0) | 1 (33.3) t279 (1) | 1 (33.3) MSSA |
| Total | | **3 (100.0)** | | **0 (0.0)** | **3 (100.0)** | **1 (20.0) MSSA = 1 (20.0) No MRSA = 0** |

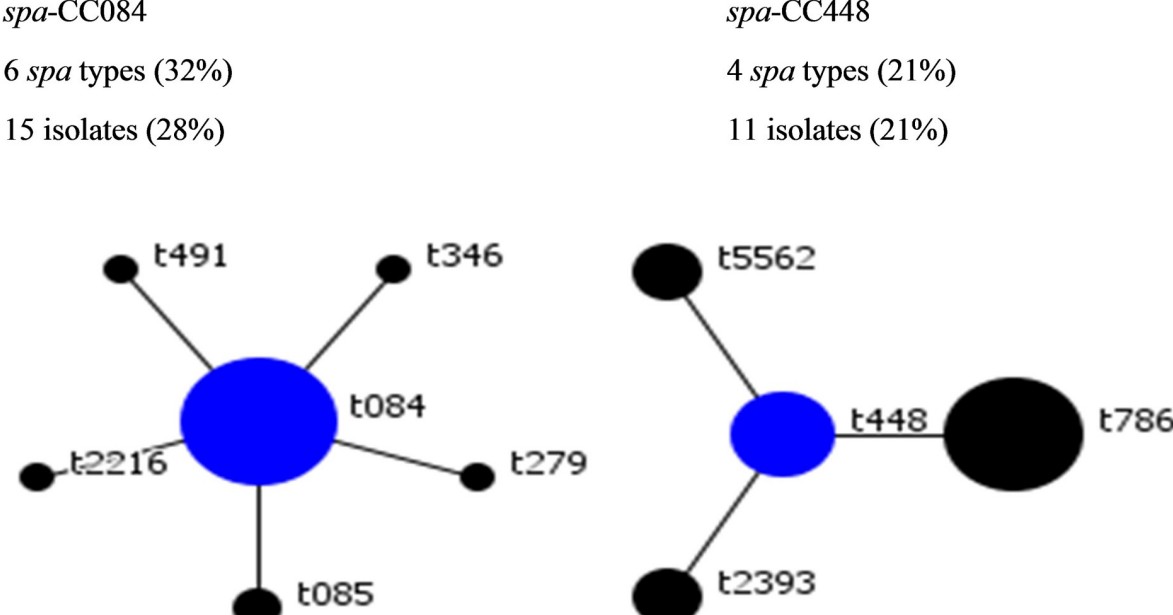

*spa*-CC084

6 *spa* types (32%)

15 isolates (28%)

*spa*-CC448

4 *spa* types (21%)

11 isolates (21%)

Singletons: t304, t311, t355, t934, t1931, t4690, t14223, t18345, t18346

**Fig 1. Based Upon Repeat Pattern (BURP) representation of *spa*-CC084 and *spa*-CC448.** ** Each node represents a *spa* type. The size of a node represents the number of isolates assigned to that *spa* type.

The 4 (36.4%) PVL-positive isolates from chicken that harbored *mecA* genes belonged to CC15: t084, t085 and CC1: t1931 (2) while the other 7 (63.6%) were MSSA and belonged to C15 (n = 4; t084), CC152 (n = 2; t355, t4690) and an undetermined CC with new *spa* type (n = 1; t18345) (Table 2).

From pigs, only 1 (20.0%) out of the 5 PVL-positive isolates harbored *mecA* genes (MRSA) and still belonged to CC1 (n = 1; t1931). The remaining 4 (80.0%) PVL-positive belonged to CC5 (n = 1; t311), CC152 (n = 1; t304) and an undetermined CC with novel *spa* type (n = 2; t18345). Two (2) PVL-positive isolates from chicken carcass handlers were MSSA in CC15 (n = 1; t084) and an undetermined CC with new *spa* type (n = 1; t18346) while 1 MSSA in CC15 (t279) was identified among isolates from pig carcass handlers (Table 2).

### Antimicrobial susceptibility of *S. aureus* isolates

The antimicrobial resistance profiles (ARPs) of 53 *S. aureus* isolates to 18 antimicrobial agents revealed that all the isolates were resistant to penicillin while 31 (58.5%) and 26 (49.1%) were resistant to sulphamethoxazole/trimethoprim and tetracycline, respectively. Resistance to erythromycin, oxacillin and clindamycin was demonstrated by 26.4%, 18.9% (only 10 out of the 12 *mecA* positive isolates were oxacillin-resistant) and 17.0% of the isolates, respectively. Levofloxacin and ciprofloxacin resistance was demonstrated by 11.3% each of the isolates while 5.7% of the isolates were resistant to gentamicin and minocycline. None of the isolates was resistant to ceftaroline, moxifloxacin, linezolid, daptomycin, vancomycin, tigecycline, nitrofurantoin and rifampicin.

The ARPs of the isolates correlated with the clonal lineages are shown in Table 3. All the isolates except for isolates belonging to CC8 exhibited MDR, with CC1 (100%) showing the

**Table 3. Correlation of the antimicrobial resistance profiles with the clonal lineages.**

| Clonal Lineage | No. of isolates | % of isolates resistant to specified antimicrobials | | | | | ARPs |
|---|---|---|---|---|---|---|---|
| | | PEN | SXT | TET | ERY | MDR | |
| CC5 | 12 | 100 | 50.0 | 16.7 | 58.3 | 66.7 | PEN,SXT,ERY |
| CC15 | 15 | 100 | 93.3 | 53.3 | 6.7 | 60.0 | PEN,SXT,TET |
| CC88 | 11 | 100 | 54.5 | 72.7 | 45.5 | 90.9 | PEN,SXT,TET,ERY |
| CC1 | 5 | 100 | 40.0 | 80.0 | 0.0 | 100 | PEN,SXT,TET |
| CC152 | 3 | 100 | 100 | 33.3 | 0.0 | 66.7 | PEN,SXT |
| CC8 | 2 | 100 | 0.0 | 100 | 0.0 | 0.0 | PEN,TET |

*PEN = penicillin, SXT = sulphamethoxazole/trimethoprim, TET = tetracycline, ERY = erythromycin, MDR = multidrug resistance, ARPs = antimicrobial resistance profiles

highest MDR, followed by CC88 (90.9%), CC5 and CC152 (66.7%), and CC15 (60.0%). TET resistance was observed for more than half of the tested isolates belonging to CC8 (100%), CC1 (80%), CC88 (72.7%) and CC15 (53.3%). SXT resistance (93.3%) was highest for isolates belonging to CC15 while ERY resistance (58.3%) was highest for CC5. However, isolates belonging to CC8 showed 100% susceptibility to SXT and ERY. Overall, CC15 and CC1 even though in different lineages showed the same major characteristic ARPs (PEN,SXT,TET) while CC5 replaced resistance to TET with ERY (PEN,SXT,ERY) and CC88 added ERY (PEN, SXT, TET,ERY).

It was observed that 64.3%, 41.2%, 40.0% and 33.3% of the isolates from chicken carcass, pig carcass, chicken carcass handlers and pig carcass handlers, respectively were multidrug resistant *S. aureus* (Table 4). Also, the *S. aureus* isolates from chicken carcass, pig carcass, chicken carcass handlers and pig carcass handlers exhibited 14, 7, 4 and 3 antimicrobial resistance patterns, respectively, with PEN-TET-SXT being the predominant in chicken carcass and PEN-ERY-SXT in pig carcasses.

## Discussion

In this study, the molecular epidemiology, genetic diversity and antimicrobial resistance of *S. aureus* in food animal carcasses and occupationally exposed individuals were investigated. *S. aureus* prevalence rates of 4.7% and 11.1% in chicken carcasses and chicken carcass handlers, and 2.8% and 6.7% in pig carcasses and pig carcass handlers, respectively, suggest that chicken meat/ carcasses and chicken carcass handlers are more contaminated with *S. aureus* than pig carcasses and pig carcass handlers. The chicken meat and handlers therefore constitute a higher risk in the transmission of *S. aureus* to the public than pig carcasses and the handlers in Enugu State, South East Nigeria. The higher prevalence of *S. aureus* in chicken carcasses and the handlers could be attributed to the increased urban poultry farming than pig farm. Humans now raise birds in the same environment where they live and so there is more contact with birds than pigs. Again, the method of processing may also be a factor contributing to the increase prevalence in chicken and the handlers, as almost all the processed chicken carcasses were washed by dipping in the same container of water. This could facilitate the contamination of chicken carcasses and the handlers unlike the pig carcasses that were washed using different containers of water because of their size. Lower isolation rates of 0.72% and 0.86% have been reported in food samples including meat from Nujiang and Yuxi provinces, respectively, in China [37]. The high level of contamination observed in this study could also be attributed to the poor state (lack of regular source of water for hygienic dressing of carcasses and cleaning the environment, no proper disposal of waste arising from the dressing carcasses, lack of

**Table 4. Antimicrobial resistance patterns of *S. aureus* isolates from food animal carcasses and occupationally exposed persons in Enugu State Southeast Nigeria.**

| Source (N) | Number of antimicrobials | Resistance pattern (Number of isolate) | Number of antimicrobial class | MDR (N, %) |
|---|---|---|---|---|
| Chicken carcass (28) | 1 | PEN(2) | 1 | 18 (64.3%) |
|  | 2 | PEN-SXT(1) | 2 |  |
|  |  | PEN-TET(1) |  |  |
|  | 3 | PEN-CIP-LEV(1) |  |  |
|  |  | PEN-MIN-TET(2) |  |  |
|  |  | PEN-OXA-TET(2) |  |  |
|  |  | PEN-TET-SXT(9) | 3 |  |
|  |  | PEN-CLI-SXT(1) | 3 |  |
|  | 4 | PEN-OXA-TET-SXT(3) | 3 |  |
|  | 4 | PEN-OXA-MIN-TET(1) | 2 |  |
|  | 4 | PEN-ERY-CLI-SXT(1) | 4 |  |
|  | 5 | PEN-OXA-GEN-MIN-TET(1) | 3 |  |
|  | 6 | PEN-CIP-LEV-ERY-CLI-TET(2) | 5 |  |
|  | 7 | PEN-CIP-GEN-LEV-ERY-CLI-TET(1) | 6 |  |
| Pig carcass (17) | 1 | PEN(4) | 1 | 7 (41.2%) |
|  | 2 | PEN-SXT(3) | 2 |  |
|  | 3 | PEN-OXA-TET(1) | 2 |  |
|  | 3 | PEN-OXA-SXT(2) | 2 |  |
|  | 3 | PEN-ERY-SXT(5) | 3 |  |
|  | 6 | PEN-CIP-LEV-ERY-CLI-TET(1) | 5 |  |
|  | 7 | PEN-CIP-GEN-LEV-ERY-CLI-TET(1) | 6 |  |
| Chicken carcass handlers (5) | 1 | PEN(2) | 1 | 2 (40.0%) |
|  | 2 | PEN-SXT(1) | 2 |  |
|  | 3 | PEN-TET-SXT(1) | 3 |  |
|  | 5 | PEN-ERY-CLI-TET-SXT(1) | 5 |  |
| Pig carcass handlers (3) | 1 | PEN(1) | 1 | 1 (33.3%) |
|  | 2 | PEN-SXT(1) | 2 |  |
|  | 3 | PEN-ERY-SXT(1) | 3 |  |

drainages for proper channeling of the waste water, flies and rodent infestations among others) of the slaughterhouses in Nigeria, especially the poultry slaughterhouses, coupled with the unhygienic practices of the uneducated butchers (open defaecation in nearby bushes, dressing of birds with very dirty clothes on dirty wooden tables, open sneezing and coughing while dressing birds among others).

In this study, 41 (77.4%) of the 53 *S. aureus* isolates studied were MSSA while 12 (22.6%) were MRSA. Although MSSA is perceived to be less virulent than MRSA, it has been reported as the most frequently-encountered bacterial pathogen in microbiology laboratories in Nigeria where it is associated with human diseases, including urinary tract infections [14, 38].

Prevalence rates of MRSA in chicken and pig carcasses were 1.5% and 0.5% respectively while no MRSA was detected among the carcass handlers. The 1.5% prevalence rate of MRSA in pig carcasses in this study contrasts the report of Momoh et al. [24] who, using the same isolation method as in this study, did not detect MRSA among *S. aureus* isolates from pigs in Jos, North-central Nigeria. The prevalence rate recorded in this study is similar to the 1.1% prevalence of MRSA in ready-to-slaughter pigs reported by Odetokun et al. [23] in Ibadan, Oyo State Southwest Nigeria. Absence of nasal carriage of MRSA by handlers of chicken and pig carcasses in this study is similar to the findings of Momoh et al. [24] among pig workers in Jos Nigeria, but it contrasted the results of other studies in Nigeria where MRSA detection ranged

from 3.4–51% among butchers and meat sellers at abattoirs and meat selling points [39–41]. Higher prevalence rates of between 43% and 84% have been reported in food animals in Nigeria [42,43] by researchers whose MRSA detection was based on phenotypic rather than the genotypic technique used in this study. This could have increased the specificity of our finding and reduced false positives.

Several studies have reported high levels of MRSA from farms in the United States of America and Europe [15,44,45]. The MRSA prevalence rates from raw retail meat products was reported to be ranging from less than 1% in Asia [46] to 11.9% in the Netherlands [47]. The reason for these higher prevalence rates than those reported in Africa and especially Nigeria is unknown and calls for further studies considering the fact that drugs are readily available and are used indiscriminately by farmers for growth promotion and prevention of diseases in livestock production.

Our data showed that 19 (35.8%) of the 53 *S. aureus* identified were PVL-positive, only 5 (26.3%) of these were MRSA. The high PVL prevalence in the MSSA isolates supports the findings of some researchers in Africa that also reported very high occurrence of PVL genes (*luk*-PV) in MSSA [14,48]. This is a consistent finding in Africa where the prevalence of PVL-positive MSSA is reported at between 17% and 74% [49], and it is a sharp contrast to what is reported in Europe and America, where the prevalence of PVL-positive MSSA is low and reported at between 0.9% and 1.4% [50].

Most of the PVL-positive MSSA isolates were obtained from chicken carcasses and chicken carcass handlers. They were classified in clonal complexes CC15, CC152 and undetermined clonal complex (CC-ND) with novel *spa* type t18345 and t18346. Okon et al. [51] reported that PVL-positive ST152 was the predominant clone in a study conducted in North-eastern Nigeria while Ruimy et al. [52] noted that it was the second most prevalent clone in a carriage study conducted in Mali, another West-African country. Bruerec et al. [49] has described the high prevalence of PVL-positive MSSA ST152 emerging in the community as well as in hospitals in West Africa. PVL-positive MSSA isolates from pig carcasses/pig carcass handlers in this study are in clonal complexes CC5, CC15 and CC152 with CC-ND: novel t18345 being the most common (2/5; 40.0%). The documentation of novel PVL-positive MSSA strains in this study suggests that new lineages of *S. aureus* capable of disseminating *luk*-PV genes have emerged in the study area.

Molecular studies such as *spa* typing and BURP algorithm enable the grouping of isolates into clonal lineages [31]. Nineteen *spa* types were detected in this study with t311 being the predominant *spa* type.

*spa* type t311 is a member of CC5. It is a common and widespread human lineage that has found its way into poultry where it is frequently being encountered and from where it has spread to other livestock [53–55]. It was recently detected in swab samples collected from the nasal cavity of farm animals (cattle and goats) and abattoir workers where these animals as well as pigs are processed [23], and in poultry in Ebonyi State, Nigeria [21] as well as from pigs in Senegal [56]. It has also been reported in clinical isolates from humans in Nigeria [14,57]. Egyir et al. [48] reported this clone as one of the predominant clones among healthcare institutions in Ghana. It is also reported as one of the predominant clones causing blood stream infections in Europe [58,59], Asia [60,61] and South America [62].

The second most common *spa* type recorded in this study, t084 (CC15), also a 'human' type clone, has been reported as one of the predominant *spa* types among isolates from food animals in Nigeria [23,24]. This *spa* type was also reported among clinical isolates of *S. aureus* in Africa [14,48,56,57,63] and Europe [64].

Detection of *S. aureus* strains in CC1, CC88 and CC152 in chicken and pig carcasses but not in persons in contact with the animal carcasses, even though they were originally human lineages, suggests the circulation of these clones in food animals only in the study area. It is

noteworthy that CC1 (t1931) harbored *luk*-PV and *mecA* genes, CC88 (t786) harbored *mecA* gene only while CC152 (t355 and t4690) harbored *luk*-PV gene only. This suggests that these clones are reservoirs of resistance (*mecA*) and virulence (*luk*-PV) genes. This is of public health concern as *mecA* gene is associated with increased antimicrobial resistance [65] while *luk*-PV gene is associated with skin and soft tissue infections and necrotizing pneumonia [66].

*spa* type t1931 (CC1) reported in this study had been detected in nasal swabs of goats and on processing/display table surfaces for sale in Southwestern Nigeria [23], and in nasal swabs of pig workers in north-central Nigeria [24]. CC1 has also been reported to be widely circulating in clinical settings in Africa [14,48], Europe [67] and Asia [68]

Momoh et al. [24] and Egyir et al. [48] also detected MSSA CC152 (t355) among isolates from pigs/pig workers and clinical samples in Nigeria and Ghana, respectively. CC152 is speculated to be widespread in West Africa but comparatively rare elsewhere unlike many other clonal complexes that are distributed worldwide [14,69].

CC88, the only clonal complex detected by Otalu et al. [25] in pigs and pig workers in Kogi State, North-central, Nigeria, also harbored *mecA* gene. CC88 has also been sporadically reported in some hospitals in Portugal [70] and Sweden [71].

Generally, CC1, CC88 and CC152 have been reported to be widely circulating in hospitals in Nigeria and Africa among immunocompromised patients [14,72]. This calls for monitoring and molecular epidemiological studies of these clonal complexes in order to know their actual sources and how to control them, since they are potential reservoirs of *luk*-PV and *mecA* genes and can be transmitted to the immunocompromised in the public through the food chain.

MSSA CC8 (t304) was detected among isolates from poultry carcasses only. Momoh et al. [24] equally detected MSSA CC8 (t304) in pigs but at a lower prevalence than reported in our study. This is a common lineage among humans [73,74] and is the origin of several MRSA clones [75]. It is one of the most prevalent clones in the United States of America [76].

In this study, 3 MSSA strains with two new *spa* types: t18345 and t18346, were detected among isolates from chicken/pig carcasses, and among persons in contact with chicken carcasses, respectively. This is a novel finding in this study and suggests that new lineages of *S. aureus* capable of disseminating *luk*-PV gene are emerging in the study area, similar to other recent findings [21,23,24].

A potential limitation of this study is that the findings cannot be used to draw a firm conclusion as to the ultimate source of the isolates in the sampled food animal carcasses and occupationally exposed individuals. This is because "human" types of *S. aureus* were found in the animal carcasses and the prototypic livestock-associated methicillin resistant *S. aureus*— CC398 was not detected in this study

The marked resistance to penicillin, tetracycline, sulphamethoxazole/trimethoprim and erythromycin is perhaps not surprising because these drugs are inexpensive, orally administered, and are available from diverse sources where they are sold with or without prescription. These antibiotics are indiscriminately used in livestock production in Nigeria [77] where backyard production of food animals is still common and where standard and hygienic farming practices are still not feasible [78]. The low levels of compliance with biosecurity practices in addition to poor husbandry practices necessitate the overdependence and indiscriminate use of these antibiotics in feed and water as growth promoters and for the prevention of diseases in poultry, piggery and other livestock production in Nigeria. Therefore, these antibiotics have found wide clinical and veterinary applications and so have been abused in Nigeria [14,77,79]. In fact, the drugs were listed in many developing countries as among the antibacterial agents that have been rendered ineffective, or for which there are serious concerns regarding bacterial resistance [14]. In pigs and poultry production in Nigeria, these drugs are commonly used for growth promotion as well as for disease prevention and control. Moreover, antimicrobial

agents sold in Nigeria and other developing countries are manufactured by combining several active ingredients of these antimicrobials at subtherapeutic/ substandard doses, so that no single drug has only one active ingredient at a required dose. Thus, lack of policies on the regulation of drug acquisition and use of cocktail of drug preparations may be contributing to the antimicrobial resistance observed in this study and other studies in Africa and Asia. Similar reports of resistance to these antibiotics have been previously reported in both human and veterinary settings as well as from retail meat products in developing countries like Ghana [48,80,81], South Africa [82] and Bangladesh [83].

Fortunately, *S. aureus* isolates in this study were highly susceptible to linezolid, daptomycin, rifampin, vancomycin, tigecycline, moxifloxacin, ceftaroline and nitrofurantoin. These are highest priority critically important antimicrobial agents in human medicine [84]. This agrees with the findings of Shittu et al. [14] who reported similar susceptibility among *S. aureus* isolated from clinical settings in Southwest, Nigeria. The high susceptibility observed could be attributed to the fact that these drugs do not have veterinary preparations and so are not available for veterinary use, and also are not routinely used in clinical setting.

Multidrug resistance was observed the most among isolates from chicken carcasses (64.3%). This indicates that chickens are more exposed to antimicrobials than pigs and humans. The *S. aureus* isolates from the chicken carcasses exhibited 14 antimicrobial resistance patterns with PEN-TET-SXT as the predominant one. Interestingly, the predominant resistance pattern was observed in one chicken carcass handler. This therefore suggests the transfer of antimicrobial resistance organisms from chicken carcasses to the chicken carcass handler. Multidrug resistance was also observed among pig carcasses (41.2%) which exhibited 7 antimicrobial resistance patterns with PEN-ERY-SXT as the predominant one. Also, the only isolate in pig carcass handlers that showed multidrug resistance exhibited a similar antimicrobial resistance pattern. This also suggests a transfer of resistance organisms to a pig carcass handler from pig carcasses.

The association of *S. aureus* antimicrobial resistance profile with their molecular characteristics and clonal lineages can provide useful information for the clinical selection of antibiotics [60]. All the clones detected in this study showed MDR except CC8. They exhibited varying degrees of resistance to tetracycline but more importantly is the high degree of resistance of CC5 to erythromycin, and C15 to sulphamethoxazole/trimethoprim. This was also reported in tertiary hospitals in China [60].

Nonetheless, the multidrug resistance results are high and therefore calls for serious concern because of the health risk associated with colonization of individuals with these MDR strains. These organisms could potentially transfer resistance genes, not only to humans in their environment but to the public/ general population, thereby jeopardizing antimicrobial therapy in carriers/infected individuals.

## Conclusion

Food animal carcasses (chicken and pig) processed as meat for human consumption in Enugu State South East, Nigeria, are potential reservoirs of PVL-producing multiple-drug resistant and methicillin-resistant *S. aureus*. This could constitute a serious public health risk. Public health intervention programs at pre- and post-slaughter stages should be considered in Nigerian slaughterhouses.

## Acknowledgments

We appreciate the kind assistance of Martha Idogwu and Mr. Ndubuisi Igwe who helped to convince the carcass handlers in their dialect to submit their nasal swabs for the study. The authors are grateful to Kimberly Yodice and Brent Christman for their assistance.

## Author Contributions

**Conceptualization:** Onyinye J. Okorie-Kanu, Madubuike U. Anyanwu, Kennedy F. Chah, Tara C. Smith.

**Data curation:** Onyinye J. Okorie-Kanu, Madubuike U. Anyanwu, Ekene V. Ezenduka, Anthony C. Mgbeahuruike, Dipendra Thapaliya, Gracen Gerbig, Ejike E. Ugwuijem, Christian O. Okorie-Kanu, Philip Agbowo, Solomon Olorunleke.

**Formal analysis:** Onyinye J. Okorie-Kanu, Ekene V. Ezenduka, Anthony C. Mgbeahuruike, Dipendra Thapaliya, Gracen Gerbig, Ejike E. Ugwuijem, Christian O. Okorie-Kanu, Philip Agbowo, Solomon Olorunleke, Kennedy F. Chah, Tara C. Smith.

**Funding acquisition:** Onyinye J. Okorie-Kanu.

**Investigation:** Onyinye J. Okorie-Kanu, Madubuike U. Anyanwu, Ekene V. Ezenduka, Dipendra Thapaliya, Ejike E. Ugwuijem, Philip Agbowo, Solomon Olorunleke.

**Methodology:** Onyinye J. Okorie-Kanu, Madubuike U. Anyanwu, Ekene V. Ezenduka, Anthony C. Mgbeahuruike, Dipendra Thapaliya, Gracen Gerbig, Ejike E. Ugwuijem, Christian O. Okorie-Kanu, Philip Agbowo, Solomon Olorunleke, John A. Nwanta, Tara C. Smith.

**Project administration:** Onyinye J. Okorie-Kanu.

**Software:** Onyinye J. Okorie-Kanu, Dipendra Thapaliya, Gracen Gerbig, Tara C. Smith.

**Supervision:** John A. Nwanta, Kennedy F. Chah, Tara C. Smith.

**Writing – original draft:** Onyinye J. Okorie-Kanu, Madubuike U. Anyanwu, Ekene V. Ezenduka, Anthony C. Mgbeahuruike, Christian O. Okorie-Kanu, John A. Nwanta, Kennedy F. Chah.

**Writing – review & editing:** Onyinye J. Okorie-Kanu, Anthony C. Mgbeahuruike, Kennedy F. Chah, Tara C. Smith.

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
