## [Decision Letter · Decision Letter 0]

17 Feb 2020

PONE-D-20-02752

Molecular epidemiology, genetic diversity and antimicrobial resistance of Staphylococcus aureus in chicken and pig carcasses and carcass handlers

PLOS ONE

Dear Dr. Okorie-Kanu,

Thank you for submitting your manuscript to PLOS ONE. After careful consideration, we feel that it has merit but does not fully meet PLOS ONE’s publication criteria as it currently stands. Therefore, we invite you to submit a revised version of the manuscript that addresses all the points raised by the reviwers during the review process.

We would appreciate receiving your revised manuscript by Apr 02 2020 11:59PM. To enhance the reproducibility of your results, we recommend that if applicable you deposit your laboratory protocols in protocols.io, where a protocol can be assigned its own identifier (DOI) such that it can be cited independently in the future. For instructions see: http://journals.plos.org/plosone/s/submission-guidelines#loc-laboratory-protocols

We look forward to receiving your revised manuscript.

Kind regards,

Herminia de Lencastre, Ph.D.

Academic Editor

PLOS ONE

Journal Requirements:

2. Your ethics statement must appear in the Methods section of your manuscript. If your ethics statement is written in any section besides the Methods, please move it to the Methods section and delete it from any other section. Please also ensure that your ethics statement is included in your manuscript, as the ethics section of your online submission will not be published alongside your manuscript.

Reviewers' comments:

Reviewer's Responses to Questions

**Comments to the Author**

1. Is the manuscript technically sound, and do the data support the conclusions?

Reviewer #1: Partly

Reviewer #2: Partly

2. Has the statistical analysis been performed appropriately and rigorously? 

Reviewer #1: No

Reviewer #2: Yes

3. Have the authors made all data underlying the findings in their manuscript fully available?

Reviewer #1: Yes

Reviewer #2: Yes

4. Is the manuscript presented in an intelligible fashion and written in standard English?

Reviewer #1: No

Reviewer #2: Yes

5. Review Comments to the Author

Reviewer #1: Comments:

1. There are sloppy mistakes in the abstract including not abbreviating Staphylococcus after the first use, even the capitalization in the title is inconsistent.

2. Inconsistent tense in lines 97-98

3. Line 162 they use t045, t002, and CC5 without any prior explanation. they need to write for a broader audience that doesn’t understand their jargon.

4. Line 187-190: I don’t think you can use Chi-square on percentages. There are much more applicable tests.

5. Line 194 don’t repeat numbers in the text that are already in Table 1.

6. Table 1: add a more explanatory description of the contents, including what is CI. How is chicken 28/600 = 0.93%. I don’t need a calculator to know that is less than 0.5%. Same for 17 out of 600 for pig

7. Line 201: presence of the MecA doesn’t necessarily mean resistant to methicillin.

8. Line 211: tell the readers why you used the spa gene! Not everyone is a Staph person. Same for the significance of the repeat successions.

9. Line 214: what is the significance of saying they were “automatically” submitted?

10. Finally on line 221 they define CC. They do list a bunch of t numbers but they have not adequately defined the importance of t numbers. I work in a different species of Staph and I don’t know what the importance of t numbers, or spa types. Thus, the impact factor of the findings is minimized. How does ANY of this relate to the global picture of S. aureus? Place Nigeria in the context of the S. aureus pan genome.

11. BURP is used many times before it is defined in the title to Figure 1. Figure 1 is very uninformative and poorly introduced. Evidently CC084 and t084 are related some how? What is the significance of the size of the circles or the colors?

12. I am perplexed why the spa typing was not related to the drug resistance profiles. That seems to be a serious under-utilization of what they have set up. They could relate spa profiles with drug resistance but they ignore it and only focus on the host source.

13. I know there is still pervasive antibiotic use in Nigeria but that is not mentioned. Was there any survey of antibiotic administration in the flocks/herds they sampled or are they all on AGP (antibiotic growth promoters)?

14. Finally at line 356 they start explaining the relevance of spa typing and BURP. But after they used it.

15. Line 361 they start to bring in a more global perspective but then dodge away.

16. They intimated that they did MLST but they did not. they used single locus typing. I actually had to go to the Ridom server and find the Home page to figure that out.

Overall: the results are worth publishing but not in the form presented. They need to go back and reanalyze their data without solely focusing on Nigeria. There is a big literature about clades of S. aureus out there. They could place the Nigeria samples in that context, without making it all just about Nigeria. They can re-work this paper and make it more readable for a wider audience and focus on the need to understand the population structure of S. aureus clades in Nigeria domestic animals and the handlers. It is not at all surprising that they can isolate pig or chicken isolates from handlers. There are numerous examples of that, but are any of them causing disease? See the PNAS paper by Ross Fitzgerald many years ago for an example.

Reviewer #2: This is a report of a cross-sectional study of S. aureus contamination of 600 chicken and 600 pig carcasses from slaughterhouses in Nigeria. There was an appropriate sampling strategy and the swabs were taken from the surface of the carcass. The chicken carcasses were handled differently (dipped in a communal water bath) than the pig carcasses which could explain the higher prevalence of contamination. In addition, they report on anterior nares colonization of 45 workers with chicken carcass contact and 45 workers with pig carcass contact. The prevalence of contamination was 4.7% in chicken carcasses, 2.8% in pig carcasses. Prevalence of anterior nares colonization was 11.0% and 6.7% respectively among the worker which is low. The low colonization rates suggests that these workers are not at increased risk because of their occupations. Spa typing, detection of the mec and PVL genes and antibiotic susceptibility was performed and analyzed appropriately. The results are presented in a very detailed fashion.

Suggestions for improving the discussions:

1. The rationale for the study is to look at animal-to-human transmission. There are overlapping spa types between workers and carcasses; however, you do not know whether the workers contaminated the carcass with SA or the carcasses transmitted SA to the workers because of the cross-sectional design. This limitation is not mentioned. There was no clear mention of what proportion of the SA isolates were livestock associated spa types; it appears from the discussion at least some were human spa types. This key point should be given more attention.

2. There was no discussion of how this compares to contamination with optimal slaughterhouse practices and what might change this contamination. The conclusion that this is an urgent public health threat is overstated.

Suggestions for improving the abstract:

1. Include the % of carcasses and workers contaminated/colonized

2. Overall #’s are distracting e.g. 1200 food animals and 90 workers- just include 600 chickens and 600 pigs and 45 chicken workers and 45 pig workers

Suggestions for Methods:

1. Include how the carcasses were swabbed – for example how much of the carcasses was sampled? How does this compare to other studies that have been done?

6. PLOS authors have the option to publish the peer review history of their article (what does this mean?). If published, this will include your full peer review and any attached files.

Reviewer #1: Yes: Douglas Duane Rhoads

Reviewer #2: No

---

## [Author Response · Author response to Decision Letter 0]

1 Apr 2020

Reviewer #1: Comments:

1. There are sloppy mistakes in the abstract including not abbreviating Staphylococcus after the first use, even the capitalization in the title is inconsistent.

Response: This was changed to all lowercase and the abstract was edited. 

2. Inconsistent tense in lines 97-98

Response: This has been edited. 

3. Line 162 they use t045, t002, and CC5 without any prior explanation. they need to write for a broader audience that doesn’t understand their jargon.

Response: This has been edited. 

4. Line 187-190: I don’t think you can use Chi-square on percentages. There are much more applicable tests.

Response: We did not use Chi-square on percentages. It was an error on our part. We have edited the statement

5. Line 194 don’t repeat numbers in the text that are already in Table 1.

Response: We removed all but the overall prevalence and significant difference between handlers and instructed readers to see Table 1. 

6. Table 1: add a more explanatory description of the contents, including what is CI. How is chicken 28/600 = 0.93%. I don’t need a calculator to know that is less than 0.5%. Same for 17 out of 600 for pig

Response: These were typographical errors and have been corrected. We included the term confidence interval to indicate CI is an abbreviation but did not define it as we expect readers will be familiar with the term. 

7. Line 201: presence of the MecA doesn’t necessarily mean resistant to methicillin.

Response: By the definition of methicillin-resistant Staphylococcus aureus (MRSA) strains, they are Staphylococcus aureus (S. aurues) strains that have an oxacillin MIC of >4µg/ml or harbor the mecA gene (Kumar et al., 2013). Chambers et al. (1997) also stated that methicillin resistance in Staphylococci is determined by mecA gene and that there is no mecA homolog in a susceptible strain of S. aureus, and so methodologies based on the detection of mecA gene are the most accurate. 

However, S. aureus isolates that carry the mecA gene but appear phenotypically methicillin/ oxacillin susceptible have been increasingly reported (Ikonomidis et al., 2008) and so it has been suggested that such isolates be classified as a new type of MRSA, designated oxacillin-susceptible methicillin-resistant S. aureus (OS-MRSA). 

Hence, Ikonomidis et al. (2008) reported that for whatever underlying reasons, the mecA gene detection and expression are prerequisites for methicillin resistance, even though such a genotype may not guarantee phenotypic methicillin resistance. However, he warned that precautions should be taken when treating OS-MRSA strains with beta-lactam antibiotics as this may result in the emergence of highly resistant MRSA, which is attributable to the presence of the mecA gene

Based on this, many researchers (Shittu et al., 2011; Thapaliya et al., 2017; Dalman et al., 2019), including us in this paper, reported and are still reporting MRSA based on the detection of the mecA gene 

8. Line 211: tell the readers why you used the spa gene! Not everyone is a Staph person. Same for the significance of the repeat successions.

Response: We have added an explanation of what the spa gene is and the importance of the repeat successions in the introduction

9. Line 214: what is the significance of saying they were “automatically” submitted?

Response: This is merely to note they have been added to the database. This has been edited slightly. 

10. Finally on line 221 they define CC. They do list a bunch of t numbers but they have not adequately defined the importance of t numbers. I work in a different species of Staph and I don’t know what the importance of t numbers, or spa types. Thus, the impact factor of the findings is minimized. How does ANY of this relate to the global picture of S. aureus? Place Nigeria in the context of the S. aureus pan genome.

Response: This is done in the discussion section, where we believe it is more appropriate. However, the spa types (t) and clonal complexes (CC) have been given prior explanation and definition in the introduction section. We have also related the spa types we got to the global picture of S. aureus, placing Nigeria in the context of the S. aureus pan genome

11. BURP is used many times before it is defined in the title to Figure 1. Figure 1 is very uninformative and poorly introduced. Evidently CC084 and t084 are related some how? What is the significance of the size of the circles or the colors?

Response: This has been edited. BURP has been defined and explained in the materials and methods. 

Figure 1 is very uninformative and poorly introduced- 

Response: This has been edited 

The significance of the size of the circle is already noted in the text: “** Each node represents a spa type. The size of a node represents the number of isolates assigned to that spa type”. We have also explained the significance of the colours

12. I am perplexed why the spa typing was not related to the drug resistance profiles. That seems to be a serious under-utilization of what they have set up. They could relate spa profiles with drug resistance but they ignore it and only focus on the host source.

Response: This has been done

13. I know there is still pervasive antibiotic use in Nigeria but that is not mentioned. Was there any survey of antibiotic administration in the flocks/herds they sampled or are they all on AGP (antibiotic growth promoters)?

Response: This was mentioned in Lines 309-402 and 405-406. Nevertheless, we added more explanations

14. Finally at line 356 they start explaining the relevance of spa typing and BURP. But after they used it. 

Response: We believe a discussion of this is most appropriate in this current location. However, we explained the relevance of spa typing in the introduction and BURP in the materials and methods, so that this would be most appropriate here 

15. Line 361 they start to bring in a more global perspective but then dodge away.

Response: We have added more global perspectives

16. They intimated that they did MLST but they did not. they used single locus typing. I actually had to go to the Ridom server and find the Home page to figure that out.

Response: We have modified it

Overall: the results are worth publishing but not in the form presented. They need to go back and reanalyze their data without solely focusing on Nigeria. There is a big literature about clades of S. aureus out there. They could place the Nigeria samples in that context, without making it all just about Nigeria. They can re-work this paper and make it more readable for a wider audience and focus on the need to understand the population structure of S. aureus clades in Nigeria domestic animals and the handlers. It is not at all surprising that they can isolate pig or chicken isolates from handlers. There are numerous examples of that, but are any of them causing disease? See the PNAS paper by Ross Fitzgerald many years ago for an example.

Response: We thank the reviewer for this comment. However, we think working to fill in the data gap in developing countries including Nigeria is a critical aspect of understanding the global epidemiology of this pathogen, and is the niche and audience we are aiming for in this publication. 

This study was not designed to analyze disease, but carriage/colonization/contamination of carcasses. We agree disease is an important aspect but is much more difficult to examine even in wealthy countries (see eg Smith & Wardyn review, Human Infections with Staphylococcus aureus CC398). We agree this would be ideally examined in future studies. 

Reviewer #2: This is a report of a cross-sectional study of S. aureus contamination of 600 chicken and 600 pig carcasses from slaughterhouses in Nigeria. There was an appropriate sampling strategy and the swabs were taken from the surface of the carcass. The chicken carcasses were handled differently (dipped in a communal water bath) than the pig carcasses which could explain the higher prevalence of contamination. In addition, they report on anterior nares colonization of 45 workers with chicken carcass contact and 45 workers with pig carcass contact. The prevalence of contamination was 4.7% in chicken carcasses, 2.8% in pig carcasses. Prevalence of anterior nares colonization was 11.0% and 6.7% respectively among the worker which is low. The low colonization rates suggests that these workers are not at increased risk because of their occupations. Spa typing, detection of the mec and PVL genes and antibiotic susceptibility was performed and analyzed appropriately. The results are presented in a very detailed fashion.

Suggestions for improving the discussions:

1. The rationale for the study is to look at animal-to-human transmission. There are overlapping spa types between workers and carcasses; however, you do not know whether the workers contaminated the carcass with SA or the carcasses transmitted SA to the workers because of the cross-sectional design. This limitation is not mentioned. There was no clear mention of what proportion of the SA isolates were livestock associated spa types; it appears from the discussion at least some were human spa types. This key point should be given more attention.

Response: This has been added

2. There was no discussion of how this compares to contamination with optimal slaughterhouse practices and what might change this contamination.

Response: This has been added

The conclusion that this is an urgent public health threat is overstated.

Response: This has been edited

Suggestions for improving the abstract:

1. Include the % of carcasses and workers contaminated/colonized

2. Overall #’s are distracting e.g. 1200 food animals and 90 workers- just include 600 chickens and 600 pigs and 45 chicken workers and 45 pig workers

Response: We have modified this. 

Suggestions for Methods:

1. Include how the carcasses were swabbed – for example how much of the carcasses was sampled? How does this compare to other studies that have been done? 

Response: This has been done

---

## [Decision Letter · Decision Letter 1]

9 Apr 2020

PONE-D-20-02752R1

Molecular epidemiology, genetic diversity and antimicrobial resistance of Staphylococcus aureus isolated from chicken and pig carcasses and carcass handlers

PLOS ONE

Dear Dr. Okorie-Kanu,

Thank you for submitting your manuscript to PLOS ONE. After careful consideration, we feel that it has merit but does not fully meet PLOS ONE’s publication criteria as it currently stands. Therefore, we invite you to submit a revised version of the manuscript that addresses the points raised by reviewer #1.

We would appreciate receiving your revised manuscript by May 24 2020 11:59PM. To enhance the reproducibility of your results, we recommend that if applicable you deposit your laboratory protocols in protocols.io, where a protocol can be assigned its own identifier (DOI) such that it can be cited independently in the future. For instructions see: http://journals.plos.org/plosone/s/submission-guidelines#loc-laboratory-protocols

We look forward to receiving your revised manuscript.

Kind regards,

Herminia de Lencastre, Ph.D.

Academic Editor

PLOS ONE

Reviewers' comments:

Reviewer's Responses to Questions

**Comments to the Author**

1. If the authors have adequately addressed your comments raised in a previous round of review and you feel that this manuscript is now acceptable for publication, you may indicate that here to bypass the “Comments to the Author” section, enter your conflict of interest statement in the “Confidential to Editor” section, and submit your "Accept" recommendation.

Reviewer #1: (No Response)

Reviewer #2: All comments have been addressed

2. Is the manuscript technically sound, and do the data support the conclusions?

Reviewer #1: Yes

Reviewer #2: Yes

3. Has the statistical analysis been performed appropriately and rigorously? 

Reviewer #1: Yes

Reviewer #2: Yes

4. Have the authors made all data underlying the findings in their manuscript fully available?

Reviewer #1: Yes

Reviewer #2: Yes

5. Is the manuscript presented in an intelligible fashion and written in standard English?

Reviewer #1: No

Reviewer #2: Yes

6. Review Comments to the Author

Reviewer #1: Review PONE-D-20-02752R1

Molecular epidemiology, genetic diversity and antimicrobial resistance of Staphylococcus aureus isolated from chicken and pig carcasses and carcass handlers

Comments to authors:

1. Need a comma after carcasses in article title

2. Line 31: Unnecessary to include ‘(S. aureus)’

3. Line 32: This study not The study

4. Line 34: include punctuation to properly associate the conjunctions. Perhaps they need a copy editor to clean up the punctuation and English.

5. Line 38, 46, 101: same problem. I will leave this issue to the editor to resolve, and not comment further.

6. Line 50-51: using abbreviations for antibiotics that are not defined.

7. Line 58 same problem as line 31

8. Line 130: purposively is not needed

9. Lines 227-228: Suggest the section title as: Clustering of isolates by spa typing and BURP analyses

10. Line 297 vs 305: antimicrobial resistance profile changes to antibiotic resistant profile and then they define an acronym. Suggest you choose one term and define early. Makes it cleaner.

11. Line 68, 201 and 307: MDR defined already

12. Line 320-321 vs 297-299: they already indicated earlier that ‘All the isolates from the four sources were resistant to penicillin’ and they seem to be repeating information in a slightly different manner in these two places. Suggest they compare these two paragraphs and remove redundancies.

13. Line 330-1: what is the point and what is the basis for the beta-lactamase screening. They just seem to be repeating numbers from the tables and they would be better just pointing to where they are drawing results from. The results still seem to just be a recitation of numbers. Just trying to encourage them to consider their readers for the importance of some of the numbers.

14. Line 334: they are back to using Staphylococcus aureus!

15. Line 366: don’t need to redefine MSSA and MRSA.

16. The bibliography needs cleaning up, to meet PLoS One format. Only some journal titles are abbreviated. I am certain the editors will catch this on acceptance.

Overall: the manuscript is far superior and highlights the importance of the work, with a focus on fitting into the global and African importance of the work. They need to clean up the coverage in the results, fix some punctuation, and get it published.

Reviewer #2: This is a report of a cross-sectional study of S. aureus contamination of 600 chicken and 600 pig carcasses from slaughterhouses in Nigeria. There was an appropriate sampling strategy and the swabs were taken from the surface of the carcass. The chicken carcasses were handled differently (dipped in a communal water bath) than the pig carcasses which could explain the higher prevalence of contamination. In addition, they report on anterior nares colonization of 45 workers with chicken carcass contact and 45 workers with pig carcass contact. The prevalence of contamination was 4.7% in chicken carcasses, 2.8% in pig carcasses. Prevalence of anterior nares colonization was 11.0% and 6.7% respectively among the worker which is low. The low colonization rates suggests that these workers are not at increased risk because of their occupations. Spa typing, detection of the mec and PVL genes and antibiotic susceptibility was performed and analyzed appropriately. The results are presented in a very detailed fashion.

All of my prior suggestions have been incorporated.

7. PLOS authors have the option to publish the peer review history of their article (what does this mean?). If published, this will include your full peer review and any attached files.

Reviewer #1: Yes: Douglas Duane Rhoads

Reviewer #2: No

---

## [Author Response · Author response to Decision Letter 1]

22 Apr 2020

We thank reviewer 1 for his thorough revisions, and all have been addressed in the revised manuscript. We really gained from his wealth of knowledge in this area. We are indeed thankful. 

We are also thankful to Reviewer 2 for his comments.

---

## [Editor Report · Decision Letter 2]

24 Apr 2020

Molecular epidemiology, genetic diversity and antimicrobial resistance of Staphylococcus aureus isolated from chicken and pig carcasses, and carcass handlers

PONE-D-20-02752R2

Dear Dr. Okorie-Kanu,

We are pleased to inform you that your manuscript has been judged scientifically suitable for publication and will be formally accepted for publication once it complies with all outstanding technical requirements.

With kind regards,

Herminia de Lencastre, Ph.D.

Academic Editor

PLOS ONE
---

## [Editor Report · Acceptance letter]

4 May 2020

PONE-D-20-02752R2 

Molecular epidemiology, genetic diversity and antimicrobial resistance of *Staphylococcus aureus* isolated from chicken and pig carcasses, and carcass handlers 

Dear Dr. Okorie-Kanu:

I am pleased to inform you that your manuscript has been deemed suitable for publication in PLOS ONE. Congratulations! Your manuscript is now with our production department. 

With kind regards,

on behalf of

Dr. Herminia de Lencastre 

Academic Editor

PLOS ONE